# Iron-catalyzed three-component amino(radio)fluorination of alkenes to unprotected β-(radio)fluoroamines

Yang Li[1,2,6], Yu Zhou[3,6], Mark R. Bortolus[4,5,6], Xiaoxuan Zhang[2], Zhitong Wang[2], Dezhi Liu[2], Hannah Le[4,5], Oaikhena Z. Esezobor[4], Jie Ni[3], Qian Zhang [ORCID][2], Neil Vasdev [ORCID][4,5], Gui-Juan Cheng [ORCID][3] ✉, Chao Zheng [ORCID][4,5] ✉ & Junkai Fu [ORCID][1,2] ✉

Unprotected β-fluoroamines are important motifs in synthetic chemistry, offering versatility for the development of β-fluorinated nitrogen-containing compounds. Existing methods to these motifs require tedious operations and suffer from low efficiencies, which has prevented their use in biologically active molecules, such as drug discovery and positron emission tomography (PET) radiotracer development. Herein, an iron-catalyzed three-component aminofluorination of alkenes using a hydroxylamine reagent and $Et_3N \cdot 3HF$ is reported, offering a direct entry to unprotected β-fluoroamines. Both aryl and unactivated alkenes are compatible, and the mild conditions along with a short reaction time enable its application in alkene aminoradiofluorination. The synthetic utility of this methodology is demonstrated by diverse follow-up derivatizations, efficient access to drug candidate LY503430, and the radio-synthesis of [$^{18}$F]KP23, a cannabinoid subtype 2 (CB2) PET radioligand. Mechanistic investigations reveal a radical pathway involving ferryl amino and aziridinium intermediates, and highlight the dual roles of $Et_3N \cdot 3HF$ as both fluorine source and reductive promotor.

The construction of β-fluoroamine motifs has attracted significant attention from organic chemists over the past few decades due to their prevalence in biologically relevant molecules, such as MK-0731 as a kinesin spindle protein (KSP) inhibitor[1], cytidine nucleoside PSI-6130 as an inhibitor of hepatitis C virus (HCV)[2], KP23 as a cannabinoid subtype 2 (CB2) receptor ligand[3], and other examples[4–6]. The incorporation of a fluorine atom into drug molecules is well known to modulate physicochemical properties, such as lipophilicity, permeability, pharmacokinetics, and metabolic stability, while also alter their structural conformation[7–11]. Moreover, the fluorine atom has been shown to attenuate the basicity of adjacent amine nitrogen atom to further improve the performance of pharmaceutical agents (Fig. 1a)[12,13]. Taking account of the versatility of aliphatic primary amines in downstream transformations to diverse secondary and tertiary amines/amides, as well as aza-heterocycles, the synthesis of unprotected β-fluoroamines is of particular interest to organic chemists[14], but also proves to be challenging since the −$NH_2$ products can strongly chelate metal catalysts and thus inhibit catalytic cycles[15]. Current synthetic methods for the synthesis of unprotected β-fluoroamines predominantly rely on sluggish nucleophilic ring-opening of

[1]National Engineering Laboratory for Druggable Gene and Protein Screening, College of Life Science, Northeast Normal University, Changchun, China. [2]Department of Chemistry, Jilin Province Key Laboratory of Organic Functional Molecular Design & Synthesis, Northeast Normal University, Changchun, China. [3]Warshel Institute for Computational Biology, School of Medicine, The Chinese University of Hong Kong, Shenzhen, Shenzhen, China. [4]Azrieli Centre for Neuro-Radiochemistry, Brain Health Imaging Centre, Campbell Family Mental Health Research Institute, Centre for Addiction and Mental Health (CAMH), Toronto, ON, Canada. [5]Department of Chemistry, Pharmacology and Toxicology, and/or Psychiatry, University of Toronto, Toronto, ON, Canada. [6]These authors contributed equally: Yang Li, Yu Zhou, Mark R. Bortolus. ✉e-mail: chengguijuan@cuhk.edu.cn; chao.zheng@camh.ca; fujk109@nenu.edu.cn

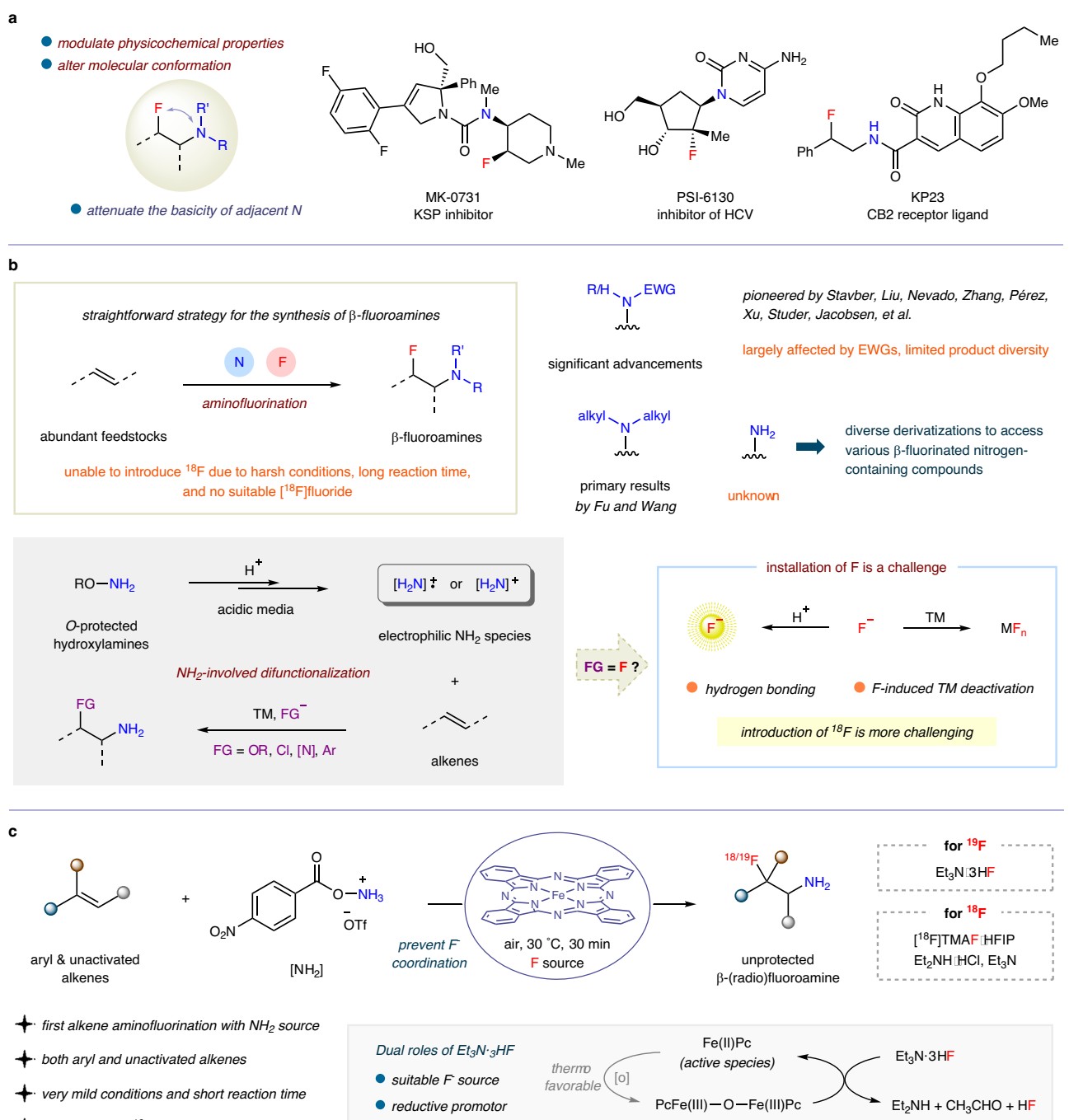

**Fig. 1 | The ubiquity and chemical synthesis of β-fluoroamines. a** Widespread presence of β-fluoroamine motifs in biologically active molecules. **b** Intermolecular aminative alkene difunctionalization in current stage. **c** Iron-catalyzed amino(radio)fluorination of alkenes to synthesize unprotected β-(radio)fluoroamines (this work).

aziridines[16,17], reduction of α-cyano/amido or β-azido fluorides[18–21], and deprotection of β-fluoro amines/amides[22–25]. However, these approaches require specially engineered precursors and suffer from low overall efficiencies.

The intermolecular aminofluorination of alkenes offers a streamlined access to β-fluoroamines by simultaneously introducing N and F atoms across abundant feedstocks (Fig. 1b)[26–28]. Considerable progress with electronically deficient amino sources has been achieved by Stavber[29], Liu[30], Nevado[31], Zhang[32], Studer[33], and among others[34–38]. These studies employed (in situ formed) electrophilic fluorinating reagents, while Pérez[39] and P. Xu[40] used nucleophilic fluorides, facilitating the synthesis of vicinal fluoroamides. A combination of electrophilic XtalFluor-E and Et₃N·3HF was utilized by H. Xu[41] to attenuate

the concentration of fluoride ion, leading to the production of carbamate-protected β-fluoroamines. But the chemical reactivities of these reactions are significantly affected by the electron-withdrawing groups (EWGs) attached to nitrogen atoms, which limits product diversity. Recently, Fu[42] and Wang[43] independently reported the alkene aminofluorination using electrophilic amination reagents as electron-rich amino sources to forge β-fluorinated tertiary alkylamines. Unfortunately, the one-step installation of a simple NH₂ group along with a fluorine atom across olefins remains unknown, but is highly desirable due to the potential for diverse derivatizations to access various β-fluorinated nitrogen-containing compounds. Additionally, because of the harsh conditions or long reaction times required, the incorporation of the radionuclide $^{18}$F with a short half-life ($t_{1/2} = 109.8$ min) and

limited availability of sources in alkene aminofluorination has not yet been achieved, significantly impeding the development of corresponding positron emission tomography (PET) radiotracers that are applied for medical imaging[44–48].

O-protected hydroxylamines have recently emerged as key precursors for $NH_2$-involved alkene difunctionalization[49], allowing the sequential introduction of various functionalities (O−, Cl−, N−, Ar−) to generate β-functionalized unprotected amines[50–63]. The major obstacle for incorporating a fluorine atom lies in the acidic media, which would likely promote the formation of highly electrophilic $NH_2$ species to facilitate alkene addition, prevent undesired rearrangement of hydroxylamine reagents to more stable N-hydroxy amides[64], and protonate the $NH_2$ unit in the product to avoid undesired chelation of the metal catalyst[15]. However, acidic conditions can also diminish the nucleophilicity of solvated fluoride ions through hydrogen bonding[65,66], and may induce competing side reactions, such as alkene protonation. Furthermore, to avoid the negative impact of a third nucleophile on aziridinium formation, one-pot/two-step procedures are often employed, as demonstrated by Lebœuf/Moran[59,60] and Ellman[61] in alkene aminoarylation and diamination, as well as Bower[62] in aminative cyclization. The highly electronegative fluoride can coordinate with transition metal complex to induce catalyst deactivation, such as iron catalyst[41], thus potentially interfering with direct radical amination processes. Considering these challenges along with its own peculiarities, the introduction of radionuclide [18]F into such reactions poses even greater technical difficulties.

Herein, we report a one-step synthesis of unprotected β-fluoroamines from simple alkenes through a three-component aminofluorination using a hydroxylamine reagent and $Et_3N·3HF$ (Fig. 1c). Phthalocyanine-coordinated iron catalyst was employed to prevent the coordination of fluoride, thus avoiding catalyst deactivation. The use of $Et_3N·3HF$ is crucial, serving both as a nucleophilic fluorine source compatible with acidic media and as a reductant that facilitated the regeneration of active ferrous species from μ-oxo diiron(III) complex to accelerate the aminofluorination reaction. This acceleration can suppress competing side reactions and lead to a very short reaction time, which, when paired with mild and air-insensitive reaction conditions, ultimately enables alkene aminoradiofluorination under the action of [18F]TMAF·HFIP, $Et_2N·HCl$, and $Et_3N$. The great synthetic potential of this methodology is fully demonstrated by broad substrate scope accommodating both aryl and unactivated alkenes, excellent functional-group tolerance, as well as diverse synthesis of β-fluorinated nitrogen-containing compounds, including the efficient synthesis of an AMPA receptor positive allosteric modulator LY503430, and unique PET radiotracer [18F]KP23 that targets the cannabinoid type 2 (CB2) receptor.

## Results
### Screening of reaction conditions
Our optimization investigations were initiated by combining 4-vinylbiphenyl **1a** with commercially available iron phthalocyanine (FePc; 0.1 equivalents) and hydroxylamine reagent $AcONH_3OTf$ (**I**; 2.5 equivalents) in acetonitrile at 40 °C under an argon atmosphere (Table 1). The choice of fluorine source played a pivotal role in the success of this reaction (entries 1–4). Substrate **1a** remained inert in the presence of basic tetramethylammonium fluoride (TMAF), while $H^+$-induced alkene decomposition and a small amount of aziridine were observed with either AgF or HF·Py, indicating a necessity of fast aminofluorination process. Delightedly, when $Et_3N·3HF$ was employed, the desired aminofluorination reaction to β-fluoroamine **2a** occurred, albeit with a low yield of 23% (Note: although the unprotected β-fluoroamine product is stable and separable, to facilitate the purification of this polar compound, a simple subsequent Boc-protection was conducted). Replacement of FePc with other types of iron salts, such as $Fe(OTf)_2$, $FeSO_4$, and $Fe(acac)_2$, resulted in complete suppression of

the reaction probably due to fluoride-induced catalyst deactivation (entries 5–7)[41]. While external ligands have been shown to improve the catalytic activity of iron catalysts[52], a combination of $FeSO_4$ with 2,2′-bipyridine or 2,2′:6′,2″-terpyridine failed in this reaction (entries 8 and 9). Further screening of reaction solvents revealed that THF gave a very low yield of only 5%, while the use of highly polar and nonnucleophilic hexafluoroisopropanol (HFIP) completely disrupted the reaction (entries 10 and 11). In contrast, reactions conducted in DCM or $CHCl_3$ provided significantly improved yields of 54% and 50%, respectively (entries 12 and 13). Optimizing the reaction temperature resulted in a modestly enhanced yield of 60% at 30 °C, and increasing the amount of $Et_3N·3HF$ from 4.0 to 8.0 equivalents showed minimal impact on the reaction outcome (entries 14–17). Subsequently, several bench-stable hydroxylamine reagents were screened (entry 18)[67]. The reactions with $PivONH_3OTf$ **II**[68,69] and $TsONH_3OTf$ **III**[70] yielded results comparable to that obtained with **I**, whereas only trace amounts of the desired product were observed with commodity chemical $HONH_3Cl$ **IV**[71]. Notably, the use of 4-$NO_2$-$BzONH_3OTf$ **V**[72] afforded **2a** in 73% yield. The reaction could be conducted open to air with only a slight reduction in yield to 71%, demonstrating its operational simplicity (entry 19, adopted as the standard reaction conditions). Extending the reaction time to 3 h did not further improve the yield, and a control experiment excluding the iron atom proved ineffective (entries 20 and 21).

### Substrate scope and derivatization
With the optimized reaction conditions in hand, we next investigated the substrate scope. As shown in Fig. 2, both styrene and its derivatives bearing either electronically neutral (−Me, −$^i$Pr, −$^t$Bu) or donating (−OMe, −OPh, −OBz) substituents were efficiently converted to products **2b**–**2j** in good yields. It is worth noting that the reaction time for **2h** was shortened to 15 min as prolonged stirring time led to gradual product decomposition, likely due to the instability of the electron-rich benzylic C−F bond under acidic conditions. Among substrates containing EWGs, F−, Cl−, Br−, and $MeO_2C$−substituted styrenes produced **2k**–**2q** in good to moderate yields, while only trace amounts of product **2r** were detected with the strongly electron-deficient −$CF_3$ group likely due to slow electrophilic addition. Moreover, both di- and sterically encumbered tri-substituted styrenes were well tolerated to furnish β-fluoroamines **2s**–**2w**. This reaction showed an excellent functional group tolerance towards chlorine atoms, azido groups, esters, C−C triple/double bonds, and even unprotected hydroxyl groups (**2x**–**2ac**), providing valuable synthetic handles for further manipulations. The 1,1-disubstituted alkenes were also found to be suitable reaction partners, and the corresponding products **2ad**–**2af** bearing tertiary carbon−fluorine stereocentres were obtained. Additionally, the aminofluorination reaction was extended to internal alkenes. In the case of cyclic internal alkene (**2ag**), an unusual cis-isomer dominated the product distribution (cis:trans = 7:1), while for acyclic internal alkene, a complete lack of stereoretention (**2ah**, a mixture containing two diastereomeric pairs in a ratio of 1:1) was observed. These results indicate a stepwise mechanism (for a detailed discussion, see Part 2 of Supplementary Information). Other arylalkenes, such as 2-vinylnaphthalene and 2-vinylbenzo[b]thiophene, were smoothly converted into the corresponding vicinal fluoroamines **2ai** and **2aj**. Importantly, this protocol was not limited to arylalkenes; unactivated alkenes were also successfully employed to deliver **2ak**–**2ao** in moderate yields with partial recovery of the starting materials. However, the reaction of mono-substituted unactivated alkene, e.g., 1-octene, failed probably due to the slow addition of N-centered radical to olefin π system that led to the decomposition of highly active radical species, and about 80% alkene substrate was recovered. To showcase the synthetic potential of this methodology in late-stage functionalization of biologically relevant molecules, the alkenes derived from natural products and pharmaceuticals, including estradiol (**2ap**), lithocholic acid (**2aq**), oxaprozin (**2ar**), ibuprofen

**Table 1 | Optimization of the reaction conditions[a]**

| Entry | [Fe] | Fluorine source | [NH₂] source | Solvent | Yield of 2a[b] |
|---|---|---|---|---|---|
| 1 | FePc | TMAF | I | CH₃CN | N.D. |
| 2 | FePc | AgF | I | CH₃CN | N.D. |
| 3 | FePc | HF·Py | I | CH₃CN | trace |
| 4 | FePc | Et₃N·3HF | I | CH₃CN | 23% |
| 5 | Fe(OTf)₂ | Et₃N·3HF | I | CH₃CN | N.D. |
| 6 | FeSO₄ | Et₃N·3HF | I | CH₃CN | N.D. |
| 7 | Fe(acac)₂ | Et₃N·3HF | I | CH₃CN | N.D. |
| 8[c] | FeSO₄ + bipyridine | Et₃N·3HF | I | CH₃CN | N.D. |
| 9[c] | FeSO₄ + terpyridine | Et₃N·3HF | I | CH₃CN | N.D. |
| 10 | FePc | Et₃N·3HF | I | THF | 5% |
| 11 | FePc | Et₃N·3HF | I | HFIP | N.D. |
| 12 | FePc | Et₃N·3HF | I | DCM | 54% |
| 13 | FePc | Et₃N·3HF | I | CHCl₃ | 50% |
| 14[d] | FePc | Et₃N·3HF | I | DCM | 60% |
| 15[e] | FePc | Et₃N·3HF | I | DCM | 45% |
| 16[d] | FePc | Et₃N·3HF (2.0 eq) | I | DCM | 43% |
| 17[d] | FePc | Et₃N·3HF (8.0 eq) | I | DCM | 61% |
| 18[d] | FePc | Et₃N·3HF | II–V | DCM | see in the graph |
| 19[d,f] | FePc | Et₃N·3HF | V | DCM | 71% (73%)[h] |
| 20[d,f,g] | FePc | Et₃N·3HF | V | DCM | 68% |
| 21[d,f] | Pc | Et₃N·3HF | V | DCM | N.D. |

[a]Reaction conditions: hydroxylamine reagent (I ~ V) (0.75 mmol) was added into a mixture of alkene 1a (0.30 mmol), [Fe] (0.03 mmol), and [F] (1.2 mmol) in DCM (2.0 mL) at 40 °C, and the reaction was allowed to proceed under N₂ for 30 min
[b]Yield of isolated product after Boc protection
[c]Addition of external ligand (0.03 mmol)
[d]At 30 °C
[e]At 50 °C
[f]Open to air
[g]For 3 h
[h]Isolated yield without Boc protection

(2as), diacetone-D-galactose (2at), phenylalanine (2au), and vanillylacetone (2av) were subjected to the optimized reaction conditions. Notably, these reactions worked well without the need for pre-protection of polar –OH or –NH groups.

The utility of this methodology for the assembly of valuable nitrogen-containing molecules is demonstrated in Fig. 3. The reaction of 2a with thiophosgene followed by H–Cl elimination furnished the β-fluorinated isothiocyanate 3. Secondary amines and amides 4−7 were synthesized via reductive alkylation, N–H insertion across an in situ formed benzyne intermediate, copper-catalyzed Ullman coupling, or dehydration condensation. Through nucleophilic substitution reactions of 2a, fluorinated tertiary amines and amides 8−10 were obtained. β-Fluoroamine 2a could also undergo Paal−Knorr reaction, four-component Debus−Radziszewski reaction, or Leuckart−Wallach/intramolecular condensation sequence to deliver various β-fluorinated aza-heterocycles, including pyrrole 11, imidazole 12, and lactam 13. Moreover, following the protection of the primary amine moiety as the corresponding amide, a subsequent Bischler−Napieralski cyclization produced dihydroisoquinoline 14,

while an intramolecular C(sp²)−H amidation/defluorination sequence afforded indole 15 (Fig. 3a). By integrating the newly developed alkene aminofluorination with a condensation reaction with thiocarbimidazole 16, an efficient route to fluorothiourea compounds 17 and 18 with potent anti-HIV activity has been established. This streamlined approach significantly simplifies the previous four-step synthetic route (Fig. 3b)[73].

The practicability of this reaction then enables a concise synthesis of LY503430 (Fig. 3c), a potential therapeutic agent for Parkinson's disease[74]. The preparation of racemic LY503430 by Eli Lilly needs more than 8 steps[75], while 14 steps are necessary for enantioselective synthesis[76]. Our approach commenced with commercially available 4'-acetyl-biphenyl-4-carboxylic acid. Amidification with methylamine followed by Wittig olefination afforded terminal alkene 20. The key alkene aminofluorination step furnished unprotected β-fluoroamine 21 in 68% yield. Final installation of a sulfonyl group onto the NH₂ moiety yielded LY503430. This route features both step economy (4 steps) and high efficiency (30% overall yield), and no tedious protection/deprotection sequence is involved.

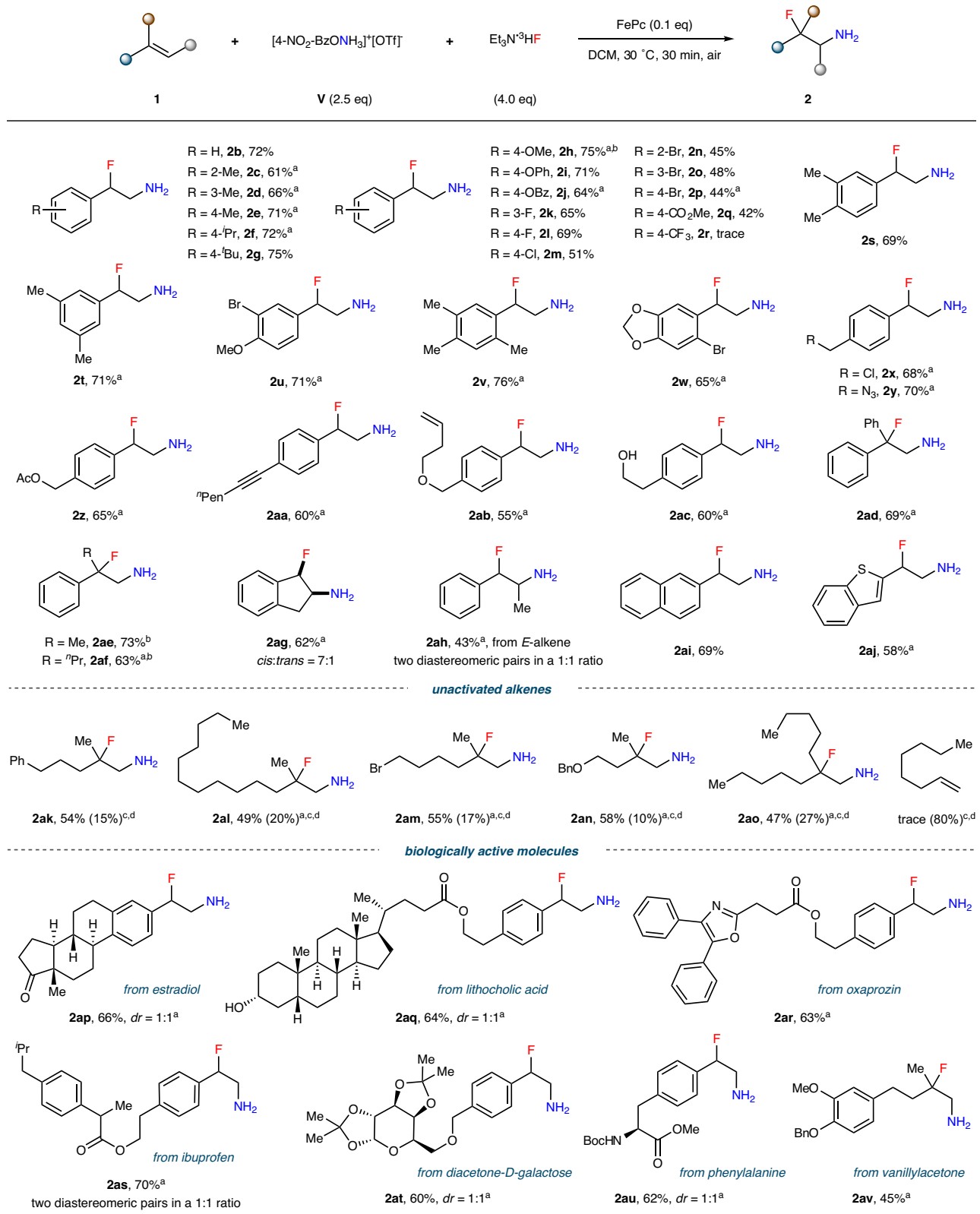

**Fig. 2 | Substrate scope.** Standard conditions: hydroxylamine reagent **V** (0.75 mmol) was added to a mixture of alkene **1** (0.30 mmol), FePc (0.03 mmol), and Et₃N·3HF (1.2 mmol) in DCM (2.0 mL) at 30 °C, and the reaction was stirred open to air for 30 min. The *drs* were determined according to [1]H NMR. [a] Yield of isolated product after Boc protection. [b]For 15 min. [c]In DCE at 80 °C. [d]Yields of recovered alkenes given in the brackets.

## Mechanistic studies

To gain insight into the underlying reaction mechanism, a series of control experiments were undertaken. The addition of 2,2,6,6-tetra-methyl-1-piperidinyloxy (TEMPO) completely suppressed the formation of β-fluoroamine **2a** (Fig. 4a). A radical clock experiment using vinylcyclopropane **22** generated ring-opening product **23** in 65% yield (Fig. 4b). These results suggested an involvement of nitrogen-centered radicals. Competition reactions were performed by treating

**Fig. 3 | Representative derivatization. a** Diverse synthesis of nitrogen-containing compounds. **b** A concise synthesis of fluorothiourea compounds. **c** Efficient access to drug candidate LY503430.

mixed alkenes (1:1) with 1.2 equivalents of hydroxylamine reagent **V**, and β-fluoroamines **2b** and **2g** dominated the reaction outcomes, respectively, indicating an electrophilic addition process strongly favoring electron-rich olefin moieties (Fig. 4c, more details in Supplementary Information). The addition of methanol led to both the

aminofluorination (**2a**) and aminooxygenation (**24**) products, with the latter likely arising from the nucleophilic ring-opening of a putative aziridinium intermediate by methanol (Fig. 4d). Furthermore, the formation of aziridinium intermediate was directly confirmed by ESI-MS analysis after conducting the reaction for just 3 min (Fig. 4e).

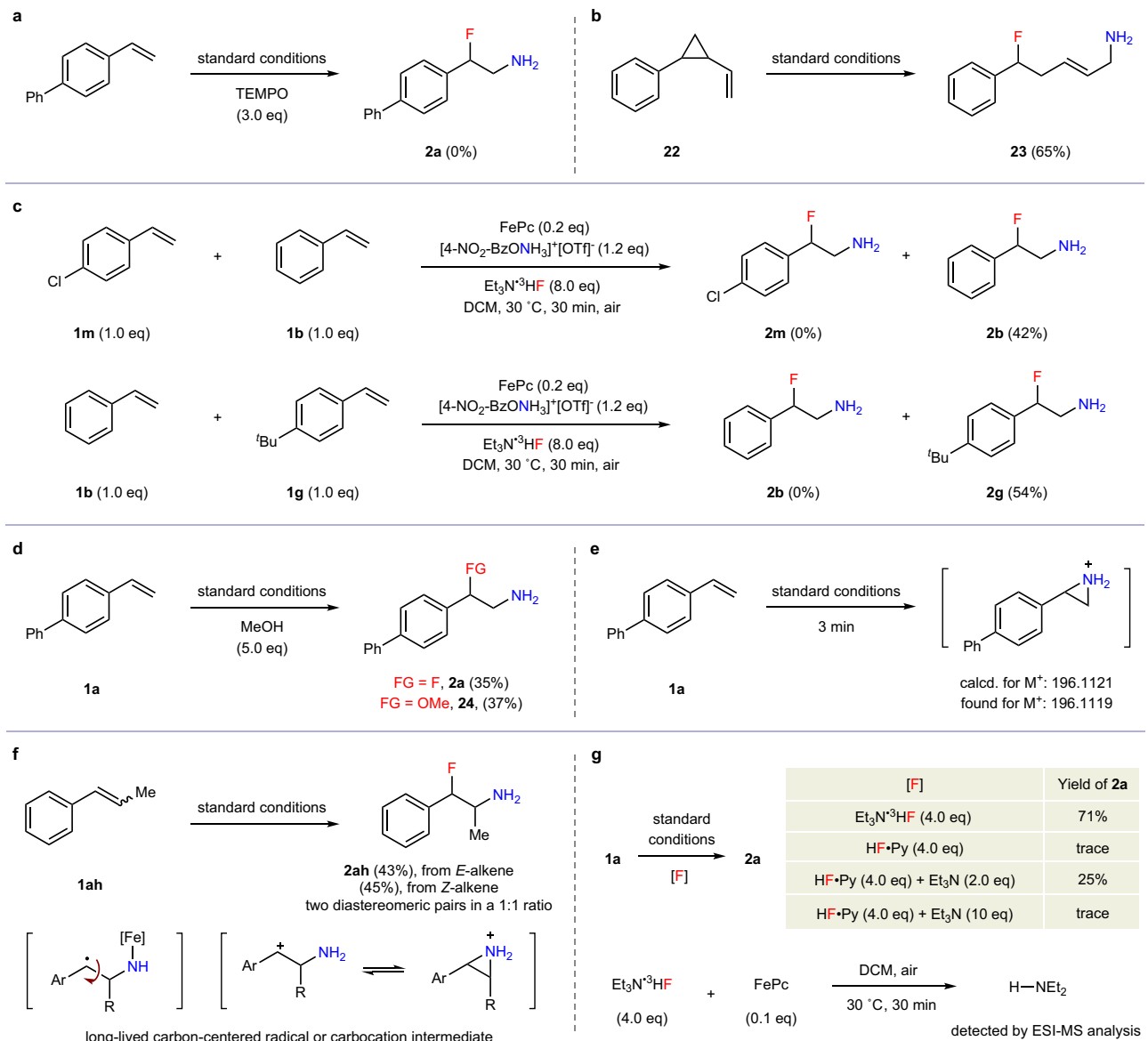

**Fig. 4 | Control experiments. a** Radical trapping experiment. **b** Radical clock experiment. **c** Competing reactions. **d** Trapping of the aziridinium intermediate. **e** Direct detection of aziridinium intermediate by ESI-MS. **f** Stereoconvergence in β-fluoroamine formation. **g** Investigation of Et₃N·3HF as a reductant.

Additionally, the use of either *E*- or *Z*-β-methyl styrene **1ah** consistently produced β-fluoroamine **2ah** as a mixture containing two diastereomeric pairs in an approximately 1:1 ratio. The complete loss of stereoinformation is indicative of a stepwise mechanism involving a long-lived carbon-centered radical or carbocation intermediate (Fig. 4f)[77]. In addition to serving as a nucleophilic fluorine source, Et₃N·3HF was found to play a special role (Fig. 4g). The addition of 2.0 equivalents of Et₃N reinitiated the aminofluorination reaction with HF·Py, offering **2a** in 25% yield; but further increasing the amount to 10 equivalents invalidated the reaction, likely due to a significant alteration in the acidity of reaction media. Moreover, after mixing Et₃N·3HF with commercially available FePc for 30 min, Et₂NH was detected by ESI-MS analysis (see Supplementary Fig. 2). These findings indicated that Et₃N·3HF might function as a reductant to accelerate the aminofluorination reaction.

To further elucidate the reaction mechanism, particularly the role of Et₃N·3HF, density functional theory (DFT) calculations have been performed. The previously proposed facile oxidation of FePc to the *μ*-oxo diiron complex ((FePc)₂O) in the presence of air is confirmed to be thermodynamically favorable (Fig. 5a)[78]. This oxidation process has

also been supported by ESI-MS, which detects the (FePc)₂O complex (see Supplementary Fig. 3). That means the commercially available FePc would be easily oxidized in part before manipulation. Additionally, the decomposition of (FePc)₂O into PcFe−OH and PcFe−F species, mediated by HF (with Et₃N·3HF simplified as Et₃N·HF)[79], is calculated to be exergonic (see Supplementary Fig. 5) and further validated by ESI-MS analysis (see Supplementary Fig. 4). However, these ferric iron species ((FePc)₂O, PcFe−OH, and PcFe−F) are found to be inactive toward the hydroxylamine reagent **V** due to the high activation barriers (>35.0 kcal/mol) for N−O bond cleavage (Fig. 5b)[80]. In contrast, computational studies reveal that the ferrous FePc complex shows significantly higher reactivity, with a low activation barrier of 16.3 kcal/mol for hydroxylamine activation (see Supplementary Fig. 6). Therefore, it is proposed that Et₃N acts as a reductant to convert these unreactive ferric iron species (PcFe−OH and PcFe−F) into the catalytically active ferrous in situ (Fig. 5c). Specifically, PcFe−F is reduced to FePc by abstracting α-hydrogen atom of Et₃N, accompanied with the formation of HF and α-amino carbon-centered radical (**INT1**). Subsequent radical rebound with PcFe−OH affords hemiaminal **INT2** and FePc[81]. Notably, alternative direct single-electron transfer from Et₃N to

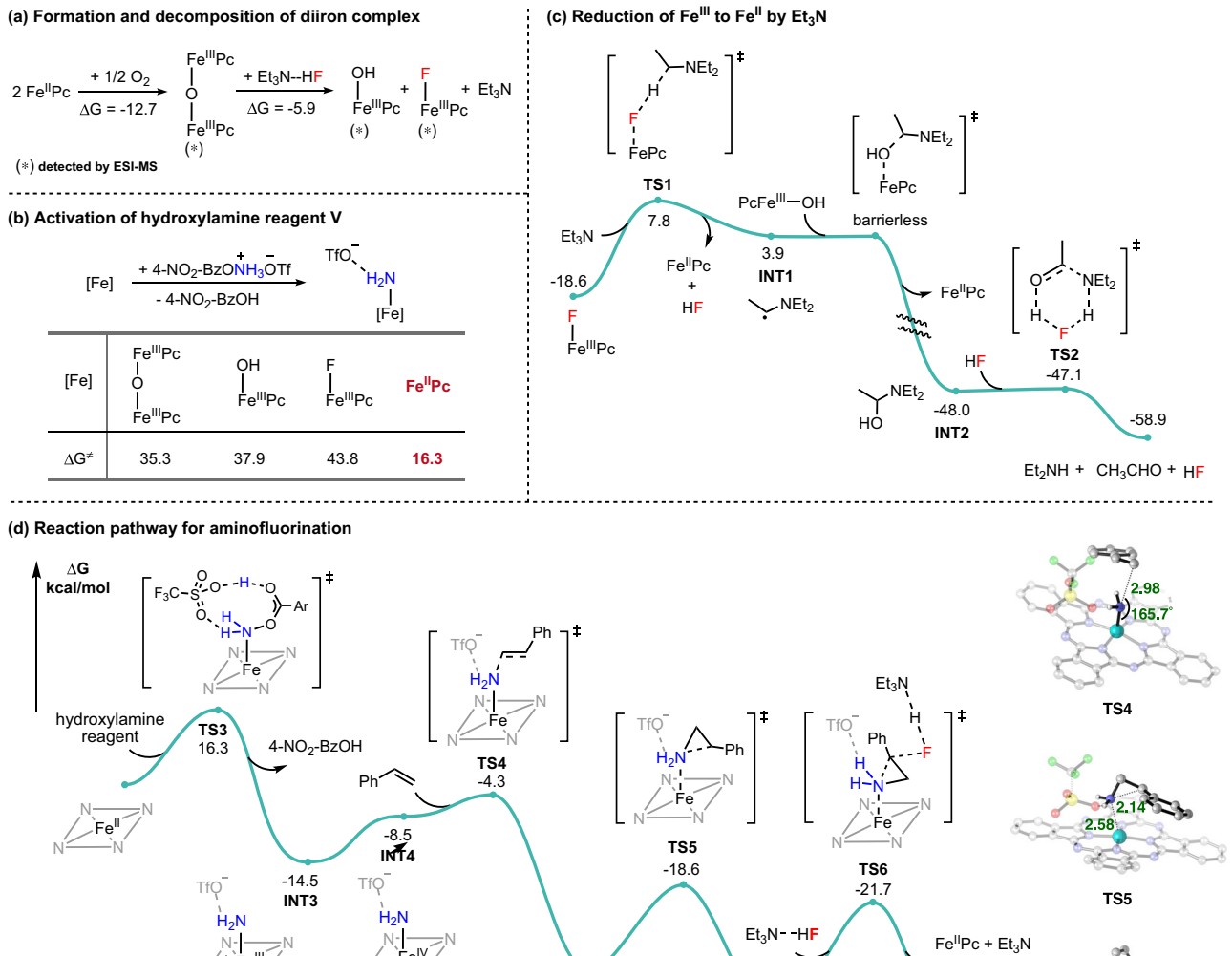

**Fig. 5 | Mechanistic investigations. a** Thermodynamic data for the formation and decomposition of diiron complex. **b** Activation of hydroxylamine reagent by different iron species. **c** Free energy profile for the reduction of Fe$^{III}$ to Fe$^{II}$ by Et$_3$N.

**d** Free energy profile for the iron-catalyzed aminofluorination of alkenes, including 3D structures for the key transition states. Energies and distances are presented in kcal/mol and angstroms (Å), respectively.

either PcFe–F or PcFe–OH is calculated to be highly endothermic (see Supplementary Fig. 7). With the assistant of HF, **INT2** readily decomposes to acetaldehyde and Et$_2$NH, the latter of which has been detected by ESI-MS (see Fig. 4g). The overall process is both kinetically feasible ($\Delta G^{\neq} = 26.4$ kcal/mol) and thermodynamically favorable ($\Delta G = -40.3$ kcal/mol). These results clearly demonstrate that Et$_3$N·3HF can significantly facilitate the reductive regeneration of the active ferrous catalyst, thus effectively minimizing side reactions and accelerating the aminofluorination reaction.

On the basis of our previous studies[82], a plausible reaction pathway is proposed in Fig. 5d. After the activation of the hydroxylamine (**TS3**), the in situ formed iron-amido intermediate (**INT3**) first experiences an electronic reconfiguration to form a ferryl amino intermediate (**INT4**), which, though less stable, is more electrophilic and somewhat distinct from typical iron-nitrenoid species[41,52,54,83–86]. This ferryl species subsequently engages in a reaction with styrene via **TS4** ($\Delta G^{\neq} = 10.2$ kcal/mol), where it receives an electron from the alkene and

reduces itself to a ferric species. The resulting benzylic radical intermediate (**INT5**) then easily undergoes cyclization to form a stable aziridinium intermediate (**INT6**) with a calculated activation barrier of $\Delta G^{\neq} = 14.1$ kcal/mol. Nucleophilic addition of Et$_3$N·3HF to **INT6** via **TS6** ($\Delta G^{\neq} = 17.5$ kcal/mol) finally affords aminofluorination product **P**. Notably, the *anti*-addition of Et$_3$N·3HF for linear alkenes, while *syn*-addition for cyclic internal alkene (consistent with **2ag** in Fig. 2) have been calculated to be favorable (see Supplementary Fig. 8). Overall, these steps are downhill processes and kinetically feasible with activation barriers less than 17.5 kcal/mol.

## Radiochemistry

Having successfully demonstrated the utility of the cold aminofluorination method, a modified synthetic protocol was developed for the direct aminoradiofluorination of alkenes to afford unprotected β-radiofluoroamines. Fluorine-18 is typically produced by bombarding $^{18}$O (oxygen-18) enriched water with protons in a cyclotron using the

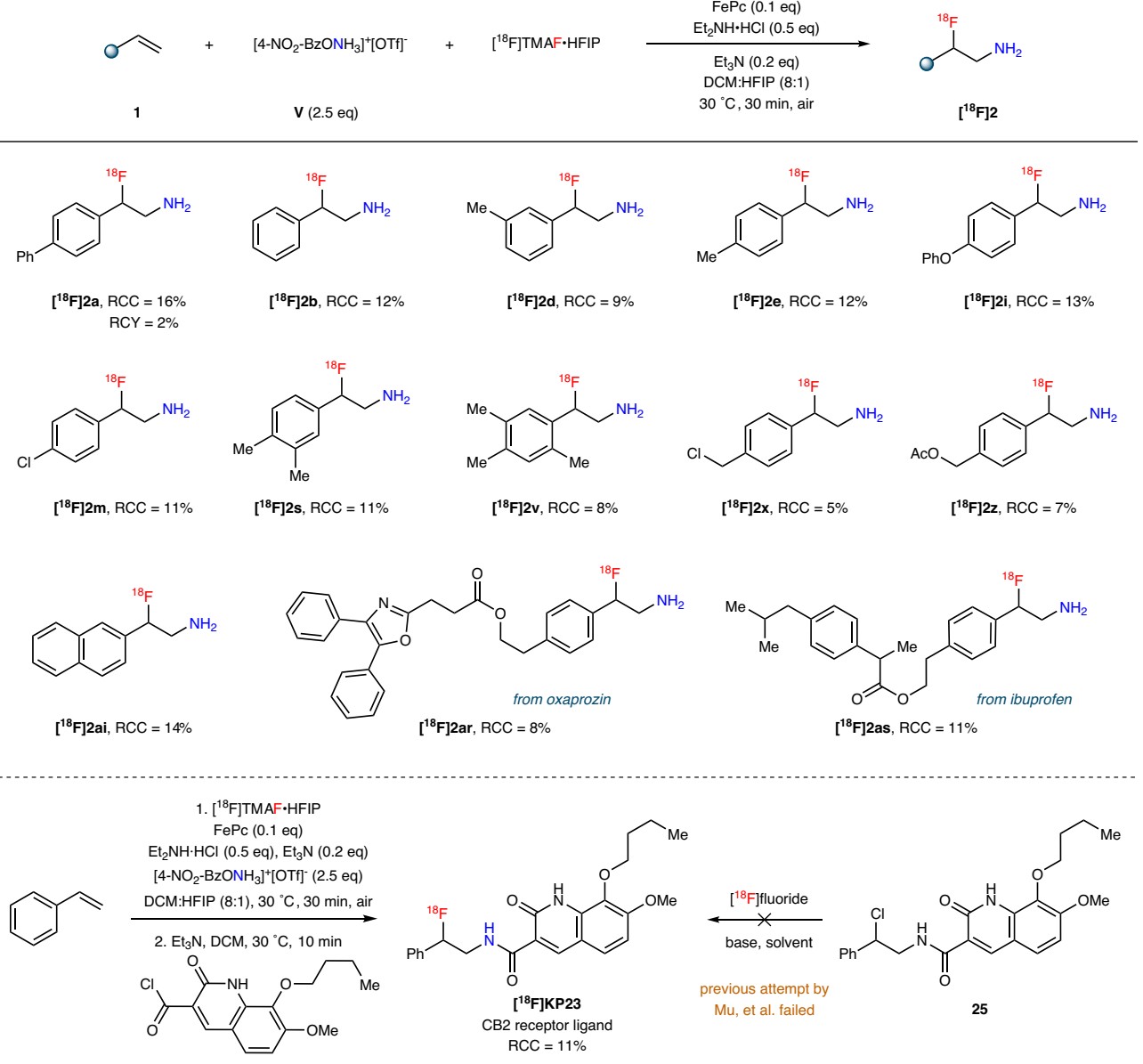

**Fig. 6 | Representative examples for the synthesis of β-radiofluoroamines.** Standard radiofluorination reaction conditions involved the addition of [18F] TMAF·HFIP and Et₃N (0.2 equiv.) in an 8:1 DCM: HFIP solution (0.4 mL) to a solid mixture containing alkene (0.025 mmol), hydroxylamine reagent **V** (2.5 equiv.), FePc (0.1 equiv.), and Et₂NH·HCl (0.25–0.5 equiv.) at 30 °C. The reaction was allowed to proceed for 30 min. Amine products were characterized and quantified as their Boc-protected derivatives by radio-TLC and/or analytic radio-HPLC. Radiochemical conversions (RCCs) determined for crude reaction mixtures are shown. A decay-corrected (d.c.) radiochemical yield (RCY) is provided for substrate [18F]**2a**.

18O(p,n)18F reaction, where the amount of 18F⁻ generated from a cyclotron is in the sub-nanomole range. This makes 18F⁻ the limiting reagent in radiochemical reactions[87–94]. Consequently, reagents containing multiple labile fluorine atoms, such as Et₃N·3HF, should be avoided to prevent isotopic dilution. After screening alternate conditions using 4-vinylbiphenyl **1a** as a model substrate (for a detailed discussion, see Supplementary Table 2), the combination of HFIP[95–97] as a hydrogen bonding agent that mediates fluoride nucleophilicity and provides a source of labile protons, TMAF as the fluoride ion source, Et₂NH·HCl as reaction additive, and Et₃N as the reductant was found to be a suitable alternative to Et₃N·3HF. Under the re-optimized cold reaction conditions, **1a** was successfully converted to **2a** and isolated as its Boc-protected derivative in 25% yield.

Manual radiochemical reaction optimization based on model substrate **1a** was then carried out (Supplementary Table 3). Comparable radiochemical conversions (RCCs) were observed at 0.05 and 0.025 mmol reaction scales, with the latter scale preferable for routine radiotracer production on automated radiosynthesis platforms. The reaction with 1.0 equivalent of Et₂NH·HCl resulted in a RCC of 10%. Reducing the loading to 0.5 or 0.25 equivalents improved the RCCs to 16% and 15%, respectively. Adjusting the reaction volume from 0.3 to 0.5 mL made little effect on the RCC, so 0.4 mL was chosen for its enhanced solubility compared to 0.3 mL at the 0.025 mmol scale. Under optimized conditions (0.025 mmol **1a**, 0.1 equiv FePc, 2.5 equiv **V**, 0.25–0.5 equiv Et₂NH·HCl, 0.2 equiv Et₃N, 0.4 mL 8:1 DCM: HFIP with 3–10 mCi [18F]TMAF·HFIP), the desired β-radiofluoroamine [18F]**2a** was obtained with a RCC of 16%. Using this method, β-radiofluoroamine [18F]**2a** was synthesized and isolated using analytical HPLC with a decay-corrected radiochemical yield (RCY)[98] of 2% and molar activity of 42.13 GBq/nmol.

A comparable substrate scope to the cold aminofluorination protocol was explored under the optimized radiofluorination

conditions. For the purposes of evaluating substrate scope, crude reaction mixtures were analyzed and their relative RCCs are discussed. All the alkene substrates were successfully labeled with moderate RCCs ranging from 5 to 16%, including those adjacent to electronically deficient, neutral, or electron-rich aromatic ring systems with a diverse array of functional groups. Di- or tri-substituted arylalkenes, as well as fused aromatic ring systems, were also well-tolerated. Complex alkene substrates derived from natural products and pharmaceuticals, such as oxaprozin and ibuprofen were able to be labeled, showcasing the viability of this method for late-stage [18F]-incorporation into drug molecules. As proof-of-concept for the (pre)clinical relevance of this method, the manual radiosynthesis of [18F]KP23 was achieved (Fig. 6). Previous attempts to prepare [18F]KP23 through nucleophilic substitution of its chloro-precursor with [18F]KF/K$_{222}$/K$_2$CO$_3$ in DMF, DMSO, or CH$_3$CN were unsuccessful[99]. This is attributed to the poor leaving group ability of the chloride in **25**. Although attempts were made to prepare tosyl- and mesyl- derivatives, these intermediates did not afford the desired KP23 due to the spontaneous displacement of the superior leaving groups by nitrogen to afford an aziridinium ion pair. This ion pair then immediately collapses in the presence of chloride-ion to reform a stable chloride derivative. In contrast, the present protocol converted commercially available styrene (**1b**) to [18F] β-fluoroamine **2b** and subsequently coupled it with acyl chloride (**12**) in a one-pot, two-step reaction without the need for intermediate purification of [18F]β-fluoroamine (**2b**), affording [18F]KP23 with a RCC of 11%. [18F]KP23 was purified through analytical HPLC with molar activity of 1.68 GBq/nmol.

## Discussion

An iron-catalyzed three-component aminofluorination of alkenes with a hydroxylamine reagent and Et$_3$N·3HF has been developed under mild reaction conditions, providing a straightforward access to unprotected β-fluoroamines from readily available feedstocks. This methodology shows a broad substrate scope, ease of operation in air, and excellent functional group tolerance, making it a practical protocol for late-stage functionalization of biologically relevant molecules. The versatility of the resulting primary amine moiety further enables the creation of structurally diverse vicinal fluorinated nitrogen-containing derivatives, including a step-economic and efficient synthesis of pharmaceutical LY503430. Detailed mechanistic investigations reveal a radical reaction pathway involving aziridinium intermediates, with Et$_3$N·3HF functioning as both a suitable nucleophilic fluorine source and a reductant that regenerates active ferrous species from μ-oxo diiron(III) complex to significantly accelerate the aminofluorination reaction. Furthermore, the dual roles of Et$_3$N·3HF can be substituted with a combination of [18F]TMAF·HFIP, Et$_2$N·HCl, and Et$_3$N, facilitating the adaptation of this methodology to radiochemistry for the direct aminoradiofluorination of a diverse range of alkene substrates. This radiochemical transformation has been employed in the radiosynthesis of fluorine-18 labeled KP23, a PET radioligand for the CB2 receptor for which no other successful synthesis had been reported. This aminofluorination approach is poised to serve as a robust framework for both drug development and advancement of PET imaging probe, offering significant potential for applications in the drug discovery and radiochemistry communities.

## Methods

### Standard reaction conditions

Hydroxylamine reagent 4-NO$_2$-BzONH$_3$OTf **V** (249 mg, 0.75 mmol, 2.5 equiv.) was added to a plastic centrifuge tube charged with alkene **1a** (54.1 mg, 0.30 mmol, 1.0 equiv.), FePc (17.0 mg, 0.03 mmol, 0.1 equiv.), Et$_3$N·3HF (196 μL, 1.2 mmol, 4.0 equiv.), and CH$_2$Cl$_2$ (2.0 mL). The mixture was stirred open to air at 30 °C (oil bath) for 30 min. Upon completion, the reaction was quenched with Et$_3$N (0.5 mL) at 0 °C. Direct purification by flash column chromatography (DCM/MeOH =

20:1 as eluent) afforded β-fluoroamine **2a** as a yellow oil (47.1 mg, 0.22 mmol, 73%).

## Data availability

The authors declare that all data supporting the findings of this research are available within the article and its Supplementary Information. Cartesian coordinates of the calculated structures are available from the Supplementary Data 1. Any further relevant data are available from the corresponding authors on request.

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

## Acknowledgements

We gratefully acknowledge the National Natural Science Foundation of China (22371036, J.F.; 21971034, J.F.; 22573090, G.C.; 22422110, G.C.; 22401041, Y.L.), Jilin Province Scientific and Technological Development Program (20230508107RC, J.F.), Shenzhen Science and Technology Program (RCYX20200714114736199, G.C.), Natural Science Foundation of Guangdong Province (2019A1515011865, G.C.), Natural Science and Engineering Research Council of Canada (RGPIN-2025-06928 and

DGECR-2025-00486, C.Z.; NSERC PDF-568046-2022, M.R.B.), and Canadian Institutes of Health Research (CIHR507113, C.Z.) for financial support. N.V. thanks the Azrieli Foundation, Canada Foundation for Innovation, Ontario Research Fund and the Canada Research Chairs Program for support.

## Author contributions

Y.L. and J.F. designed and performed the experiments. X.Z., Z.W. and D.L. assisted in completing the experiments. Y.Z. and G.-J.C performed the density functional theory calculations and wrote the sections for the manuscript. J.N. performed the ESI-MS experiments to capture the intermediates. M.R.B., H.L., O.Z.E., N.V. and C.Z. designed and carried out all radiochemical experiments and wrote the radiochemical sections of the manuscript. Q.Z., C.Z. and J.F. directed the project and wrote the manuscript. All the authors were involved in the interpretation of the results presented in the manuscript.

## Competing interests

The authors declare no competing interests.
