## [Transparent Peer Review File · Nature Communications]

Iron-Catalyzed Three-Component Amino(radio)fluorination of Alkenes to Unprotected β -(radio)Fluoroamines

Corresponding Author: Dr Chao Zheng

Version 0:

Reviewer comments:

Reviewer #1

(Remarks to the Author)

In the submitted manuscript, Fu, Zheng, and Cheng developed an appealing iron-catalyzed three-component aminofluorination reaction by mixing simple alkenes with hydroxylamine salt and commercially available nucleophilic F reagent, giving rise to the corresponding β -fluoroamines bearing a free NH₂ group. This work greatly enriches the practicability of alkene aminofluorination, extending the amino sources from previous amides and secondary amines to synthetically flexible primary amines. This advance allows for a collective synthesis of structurally diverse vicinal fluorinated nitrogen-containing compounds and an efficient access to drug candidate LY503430. The mechanistic studies are comprehensive and impressed, illustrating the plausible reaction pathway and highlighting the dual role of Et₃N-HF as both a reductant and the F⁻ anion source. Moreover, the mild reaction conditions at room temperature and a fast reaction time in 30 min enable a first-in-class amino(radio)fluorination method for unprotected β -(radio)fluoroamines, which I think is a breakthrough in alkene aminofluorination. Overall, the reviewer considers that this manuscript should be accepted after addressing a few minor suggestions.

1) In the introduction part, the sentence "By contrast, the straightforward installation of a simple NH₂ group along with a fluorine atom across olefins remains a formidable challenge in organic synthesis..." is not accurate, without literature to support. A revised "Unfortunately, the one-step installation of a simple NH₂ group along with a fluorine atom across olefins remains unknown..." should be better.

2) A recent work reported by Bower on alkene 1,2-aminohydroxylation should be cited in the main text (Angew. Chem. Int. Ed. 2024, 63, e202409836).

3) In Figure 2, do the mono-substituted terminal unactivated alkenes work well for the aminofluorination reaction? The authors are encouraged to give the primary results.

4) In Figure 2, the result for 2ac shows a diastereomeric ratio (dr) of 7:1 rather than 1:1. What is the reason for this discrepancy?

5) In Figure 6, the radiochemical yield (RCY) of compound KP23 is labeled as "d.c." Are all the radiochemical yields decay-corrected? Please clarify "d.c." when it was first mentioned in the main text.

6) What is the role of HFIP in the amino(radio)fluorination, and how does it contribute to increasing the reaction yield? Regarding to HFIP, the following papers should be cited: Org. Lett. 2024, 26, 3435. ACIE. 2018, 57, 22,2924.

7) Could the authors explain why the previous synthesis of ¹⁸F-KP23 failed using traditional methods, and suggest potential reasons for this?

Reviewer #2

(Remarks to the Author)

The article proposal by Gui-Juan Cheng, Chao Zheng, Junkai Fu, and co-workers presents a method for accessing β -fluoroamines from alkenes through an operationally simple process. The manuscript provides a broad reaction scope, including examples of biologically relevant substrates and amine derivatizations. Mechanistic investigations and applications to ¹⁸F radiochemistry are also explored. Despite the results obtained, I believe this article is not suitable for Nature Communications for several reasons outlined below:

1) This work is (too) inspired by Morandi's 2018 Science publication on the synthesis of parent β -chloroamines. The main advancements lie in identifying the appropriate fluorine source and iron catalyst. However, the FePC catalyst employed by

the authors was previously reported by Morandi in a separate study (Angewandte 2016) on amino-alcohol synthesis. As a result, I find this work lacking in originality.

2) The article title suggests a transformation from alkenes to unprotected β -fluoroamines. However, upon closer examination of the manuscript, only two unprotected β -fluoroamines (2a and 2b) are directly synthesized, while the majority of the reported products are Boc-protected derivatives. A double numbering system is used, and the reported yields for the amines actually correspond to those of the Boc derivatives. Furthermore, I found no evidence of Boc deprotection, either in the manuscript or the supporting information. The NH_2 compounds 2a and 2b are the only examples of substrates used in derivatization examples.

3) The reaction scope for styrene derivatives primarily focuses on electron-donating substituents, while only a few electron-withdrawing groups are explored, and always in the same position. However, this criterion is initially claimed in the article as a distinguishing feature compared to examples of β -fluoroamine synthesis using electrophilic fluorination reagents.

4) The scope for unactivated alkenes shows yields around 50%. Could this indicate a selectivity issue? This point should be discussed.

5) The description of diastereomeric ratios is problematic for compounds 2ad and 2ao (Figure 2), where two pairs of diastereomers are present in the mixture. The methodology used to determine the other diastereomeric ratios should be clearly explained. Additionally, compound 2ac (Figure 2) is misrepresented, as the actual product is a mixture of enantiomers.

6) The discussion on ^{18}F radiochemistry is problematic. Radiochemical yield (RCY) has a precise definition (Nucl. Med. Biol. 2021, 93, 19–21), which requires the ^{18}F -labeled derivative to be purified, isolated, and its activity measured. The reported RCYs values appear to be an extrapolations from radiochemical conversions (RCC) and may not accurately reflect reality. Furthermore, at least one correctly assessed RCY should be reported, along with a specific activity measurement, to determine whether this approach is viable for PET tracer production—the primary goal of ^{18}F radiochemistry.

Reviewer #3

(Remarks to the Author)

In this work, the authors described a FePc-catalyzed aminofluorination of alkenes, providing a valuable route to N-protected β -fluoroamines. Although there have been some pioneering reports on alkene aminofluorination reactions (ref. 26-43), and studies on the use of hydroxylamine reagents as NH_2 radical sources in alkene difunctionalizations (ref. 50-62), this work remains noteworthy, as it offers an efficient method for preparing synthetically valuable β -fluoroamines, without the need for deprotection steps. The application in radiosynthesis further highlighted the practical utility of this work. The authors also demonstrate a range of synthetic transformations of the reaction products, including the synthesis of several pharmaceutically relevant molecules. Mechanistic studies, including DFT calculations, were performed, revealing a potential catalytic pathway for this reaction. Overall, the manuscript is well-organized, and I believe it would be of interest to the readership of Nat. Comm., after addressing the following problems/issues:

1. The authors state that unactivated alkenes are compatible with this reaction, yet all the examples in Figure 2 are substituted styrenes or 1,1-disubstituted alkenes. The reactivity of monosubstituted alkenes such as 1-octene should be discussed, and an example should be provided if possible.

2. In DFT calculations, the oxidation state of Fe in INT4 is described as +4, which is somewhat confusing, as such a formal +4 oxidation state is typically exhibited only in iron-nitrenoid species. A more precise description of this species should be included in the manuscript. Additionally, is it possible for the reaction to proceed directly from INT3 without undergoing INT4?

3. In Figure 5, is there a hydrogen bond between aziridinium and FePc in INT6? The structural formula and description are unclear and may be misleading. Moreover, another potential mechanistic pathway should be considered, in which FePc has already dissociated in INT6, and TS6 corresponds to a nucleophilic addition of $\text{Et}_3\text{N}\cdot 3\text{HF}$ to the aziridinium without Fe-catalysis. Additionally, I could not find the Cartesian coordinate of TS6 in the SI, and it seems that the author may have mixed up the numbering. A careful review of the SI is necessary.

4. Some typos:

Line 138, reepectively -> respectively

Line 200, soure -> source.

Version 1:

Reviewer comments:

Reviewer #1

(Remarks to the Author)

I have reviewed the authors' responses and revisions addressing my comments. They have thoroughly and satisfactorily addressed all points raised. With these revisions, the manuscript is significantly strengthened and fully suitable for acceptance in its current form.

Reviewer #2

(Remarks to the Author)

I read with great interest the modifications, the comments from the other reviewers, as well as the authors' responses to all reviewers. Overall, I believe the substantial effort made by the authors in addressing the comments brings this article closer

to acceptance, aside from a few remaining details particularly regarding two of my remarks.

Point number 2)

The authors have shown that they can obtain amines directly for a large number of derivatives. However, I still find Figure 2 problematic in terms of the molecule numbering. It is not acceptable to display molecules in this figure with a “2x” number and an associated yield, represented as amines, while in the Supporting Information these same molecules are in fact carbamates with the same yield. I find the footnote system used in Figure 2 (a and b) by the authors to be lacking in rigor, and I believe this needs to be corrected. The same issue appears in Figure 6 for the radiolabeled products.

Point number 6)

The authors have clarified the misunderstandings regarding RCC and RCY and have added two molar activities. However, they should report the specific activities using a more conventional unit, GBq/ μ mol for example. While the reported values are good, using a more appropriate unit would allow readers to understand them at a glance (EJNMMI Radiopharm. Chem. 6, 34 (2021). <https://doi.org/10.1186/s41181-021-00149-6>). Additionally, the authors should, in my opinion, include in the Supporting Information the data used to determine the reported molar activities (calibration curve, total and isolated activities....). I believe this is consistent with current standards in radiochemistry (J. Am. Chem. Soc. 2024, 146, 11599–11604).

Reviewer #3

(Remarks to the Author)

All of my concerns have been well addressed. I recommend accepting this manuscript for publication in Nature Communications.

Response to the Reviewer 1's comments

In the submitted manuscript, Fu, Zheng, and Cheng developed an appealing iron-catalyzed three-component aminofluorination reaction by mixing simple alkenes with hydroxylamine salt and commercially available nucleophilic F reagent, giving rise to the corresponding β -fluoroamines bearing a free NH₂ group. This work greatly enriches the practicability of alkene aminofluorination, extending the amino sources from previous amides and secondary amines to synthetically flexible primary amines. This advance allows for a collective synthesis of structurally diverse vicinal fluorinated nitrogen-containing compounds and an efficient access to drug candidate LY503430. The mechanistic studies are comprehensive and impressed, illustrating the plausible reaction pathway and highlighting the dual role of Et₃N-HF as both a reductant and the F anion source. Moreover, the mild reaction conditions at room temperature and a fast reaction time in 30 min enable a first-in-class amino(radio)fluorination method for unprotected β -(radio)fluoroamines, which I think is a breakthrough in alkene aminofluorination. Overall, the reviewer considers that this manuscript should be accepted after addressing a few minor suggestions.

Response: We thank the Reviewer for the positive evaluation and strong support of this work.

1) In the introduction part, the sentence “By contrast, the straightforward installation of a simple NH₂ group along with a fluorine atom across olefins remains a formidable challenge in organic synthesis...” is not accurate, without literature to support. A revised “Unfortunately, the one-step installation of a simple NH₂ group along with a fluorine atom across olefins remains unknown...” should be better.

Response: We have made the revision according to the valuable suggestion.

2) A recent work reported by Bower on alkene 1,2-aminohydroxylation should be cited in the main text (*Angew. Chem. Int. Ed.* 2024, 63, e202409836).

Response: According to the suggestion of the Reviewer, Bower's recent work (*Angew. Chem. Int. Ed.* 2024, 63, e202409836) has been cited as ref. 63 in the revised manuscript.

3) In Figure 2, do the mono-substituted terminal unactivated alkenes work well for the aminofluorination reaction? The authors are encouraged to give the primary results.

Response: According to the suggestion of the Reviewer, we have employed 1-octene as the substrate. Unfortunately, the reaction failed probably due to the slow addition of *N*-centered radical to the π system of mono-substituted unactivated alkene, which led to the decomposition of highly active radical species. About 80% alkene substrate was recovered. This information has been added in the revised main text and Figure 2.

4) In Figure 2, the result for 2ac shows a diastereomeric ratio (dr) of 7:1 rather than 1:1. What is the reason for this discrepancy?

Response: These results can be explained through a stepwise mechanism. As shown below, after the addition of *N*-centered radical onto the acyclic internal alkene, the rotatable C–C bond allows for the formation of two aziridinium ion isomers. An *anti*-addition of fluoride ion to the

aziridinium ion species then produces the β -fluoroamine products as a mixture containing two diastereomeric pairs in a ratio of 1:1.

On the other hand, for cyclic internal alkene, the conformation limitation will prohibit the free rotation of the C–C bond for carbon radical intermediate. This leads to the formation of single aziridinium ion intermediate, which undergoes *syn*-addition by fluoride ion to finally produce the *cis*- β -fluoroamine as the major product (*cis:trans* = 7:1).

The detailed discussion on selective control has been added in the revised SI as Part 2.

5) In Figure 6, the radiochemical yield (RCY) of compound KP23 is labeled as "d.c." Are all the radiochemical yields decay-corrected? Please clarify "d.c." when it was first mentioned in the main text.

Response: We have made the revision to define decay-corrected at first-mention in the text.

6) What is the role of HFIP in the amino(radio)fluorination, and how does it contribute to increasing the reaction yield? Regarding to HFIP, the following papers should be cited: *Org. Lett.* 2024, 26, 3435. *ACIE.* 2018, 57, 22, 2924.

Response: HFIP serves as a hydrogen bonding agent which mediates fluoride nucleophilicity and provides a source of labile protons. An explanation to this effect has been included in the revised manuscript. The references provided have been included in ref. 95-97, as well as a recent publication from Gouverneur and co-workers which uses HFIP as an additive in a radiofluorination method.

7) Could the authors explain why the previous synthesis of ^{18}F -KP23 failed using traditional methods, and suggest potential reasons for this?

Response: The explanation provided by the authors of the original paper is summarized and included in the relevant section of the revised manuscript as follows:

“This is attributed to the poor leaving group ability of the chloride in **25**. Although attempts were made to prepare tosyl- and mesyl- derivatives, these intermediates did not afford the desired KP23 due to the spontaneous displacement of the superior leaving groups by nitrogen to afford an aziridinium ion pair. This ion pair then immediately collapses in the presence of chloride-ion to reform a stable chloride derivative.”

Response to the Reviewer 2's comments

The article proposal by Gui-Juan Cheng, Chao Zheng, Junkai Fu, and co-workers presents a method for accessing β -fluoroamines from alkenes through an operationally simple process. The manuscript provides a broad reaction scope, including examples of biologically relevant substrates and amine derivatizations. Mechanistic investigations and applications to ^{18}F radiochemistry are also explored. Despite the results obtained, I believe this article is not suitable for *Nature Communications* for several reasons outlined below:

1) This work is (too) inspired by Morandi's 2018 *Science* publication on the synthesis of parent β -chloroamines. The main advancements lie in identifying the appropriate fluorine source and iron catalyst. However, the FePC catalyst employed by the authors was previously reported by Morandi in a separate study (*Angewandte* 2016) on amino-alcohol synthesis. As a result, I find this work lacking in originality.

Response: Alkene difunctionalization using *O*-protected hydroxylamines provides an efficient route to β -functionalized unprotected amines. Following the pioneering work by Morandi (*Angew. Chem. Int. Ed.* **2016**, *55*, 2248; *Science* **2018**, *362*, 434), the groups of Falck, Arnold, Che, Leboeuf/Moran, and Ellman make significant contributions to this chemistry, with *O*-, *Cl*-, *N*-, and *Ar*-based functionalities have been successively introduced (see ref. 50-63 in the revised manuscript). However, the incorporation of F atom remains unknown. Herein, we report a FePc-catalyzed three-component aminofluorination of alkenes with hydroxylamine reagent and $\text{Et}_3\text{N}\cdot 3\text{HF}$, providing a one-step entry to unprotected β -fluoroamines. The innovative aspects of this work are highlighted below:

- (1) This methodology represents the first example of a NH_2 -involved alkene aminofluorination reaction, expanding the boundary of this chemistry. The resulting unprotected β -fluoroamines allow for convenient derivatizations to a collective synthesis of structurally diverse β -fluorinated nitrogen-containing compounds that previously being inaccessible or requiring elaborate synthetic efforts (as shown in Figure 3).
- (2) Mechanistic studies herein will guide future development of aminofluorination reactions. The FePC catalyst is found to suppress fluoride-induced catalyst deactivation, while a dual role of $\text{Et}_3\text{N}\cdot 3\text{HF}$ was identified as both a nucleophilic fluorine source compatible with acidic media and a reductant facilitating the regeneration of active ferrous species is crucial for the success of this aminofluorination reaction.
- (3) The mild and air-insensitive conditions, along with a very short reaction time contribute to the first introduction of radionuclide ^{18}F in alkene aminofluorination.

In view of the significant potential of this methodology for applications in the organic chemistry, drug discovery, and radiochemistry communities, we believe that this work will be of broad interest to the readership of *Nature Communications*.

2) The article title suggests a transformation from alkenes to unprotected β -fluoroamines. However, upon closer examination of the manuscript, only two unprotected β -fluoroamines (2a and 2b) are directly synthesized, while the majority of the reported products are Boc-protected derivatives. A double numbering system is used, and the reported yields for the amines actually correspond to those of the Boc derivatives. Furthermore, I found no evidence of Boc deprotection,

either in the manuscript or the supporting information. The NH₂ compounds **2a** and **2b** are the only examples of substrates used in derivatization examples.

Response: The subsequent Boc-protection of NH₂ group is to facilitate the purification of the aminofluorination products. According to the suggestion of the Reviewer, several representative alkene substrates are selected to perform the aminofluorination reaction without additional Boc-protection step, including styrene derivatives bearing electronically neutral (**2b**, **2g**), donating (**2i**), and withdrawing (**2k-2o**, **2q**) substituents, di-substituted styrene (**2s**), 1,1-disubstituted styrene (**2ae**), 2-vinylnaphthalene (**2ai**), and unactivated alkene (**2ak**). The desired unprotected β -fluoroamine products were isolated according to the general procedure A using DCM/MeOH as eluent. The corresponding experimental details and NMR spectra have now been added in the revised SI.

Meanwhile, we have tested the removal of the Boc group according to the suggestion of the Reviewer. For example, treatment of Boc-protected compound **2a'** with trifluoroacetic acid in DCM at 0 °C for 1 h produced unprotected β -fluoroamine **2a** in 89% isolated yield. The experimental details have been added in Part 6 of the revised SI.

3) The reaction scope for styrene derivatives primarily focuses on electron-donating substituents, while only a few electron-withdrawing groups are explored, and always in the same position. However, this criterion is initially claimed in the article as a distinguishing feature compared to examples of β -fluoroamine synthesis using electrophilic fluorination reagents.

Response: In light of the Reviewer's comments, we have now performed the aminofluorination reaction with more styrene derivatives bearing electron-deficient substituents. In addition to 4-F-, 4-Cl-, and 4-Br-substituted styrenes (**2l**, **2m**, and **2p**), the reactions of 2-F-, 2-Br-, and 3-Br-substituted styrenes all worked well to give corresponding unprotected β -fluoroamines **2k**,

2n, and **2o**. Moreover, the styrene substituted with an electron-deficient ester group underwent the aminofluorination reaction to furnish the desired product **2q** in moderate yield. These results showcase the broad applicability of this methodology, and have been added into Figure 2 in the revised manuscript.

4) The scope for unactivated alkenes shows yields around 50%. Could this indicate a selectivity issue? This point should be discussed.

Response: According to the suggestion of the Reviewer, we reanalyzed the reaction outcomes of unactivated alkenes. Besides the desired aminofluorination products, approximately 20% of the starting alkene substrates (given in the bracket) remained unreacted. Prolonging the reaction time or adding more equivalents of hydroxylamine reagent did not promote the consumption of the remaining alkenes. No other selective isomers were detected. This information has been added in Figure 2 of the revised manuscript.

5) The description of diastereomeric ratios is problematic for compounds **2ad** and **2ao** (Figure 2), where two pairs of diastereomers are present in the mixture. The methodology used to determine the other diastereomeric ratios should be clearly explained. Additionally, compound **2ac** (Figure 2) is misrepresented, as the actual product is a mixture of enantiomers.

Response: We thank the Reviewer for this helpful suggestion. We agree with this point, and the standard *dr* may not suffice because there are four stereoisomers (two diastereomeric pairs) instead of just two. In this case, we describe the products **2ah** and **2as** (previous **2ad** and **2ao**) as a mixture containing two diastereomeric pairs in a ratio of 1:1. For product **2ag** (previous **2ac**), using *cis:trans* may be better compared to *dr*. The other diastereomeric ratios (*drs*) were determined according to ¹H NMR.

Accordingly, we have made a revision in Figure 2 and Figure 4f to clarify these points.

6) The discussion on ^{18}F radiochemistry is problematic. Radiochemical yield (RCY) has a precise definition (Nucl. Med. Biol. 2021, 93, 19–21), which requires the ^{18}F -labeled derivative to be purified, isolated, and its activity measured. The reported RCYs values appear to be an extrapolations from radiochemical conversions (RCC) and may not accurately reflect reality. Furthermore, at least one correctly assessed RCY should be reported, along with a specific activity measurement, to determine whether this approach is viable for PET tracer production—the primary goal of ^{18}F radiochemistry.

Response: We agree with the Reviewer that RCY and RCC values are often misrepresented in the literature, and RCY as it pertains to the decay corrected yield determined from a radio-TLC or radio-HPLC analysis of a crude reaction mixture (product is not isolated, RCY is extracted from RCC) is utilized in recent radiochemical papers published in high impact journals (*e.g.*, Ortalli, S., et al, *J. Am. Chem. Soc.* **2024**, *146*, 11599-11604). We respectfully acknowledge that such publications are misaligned from the harmonized RCY definition used for PET radiotracer production as radiochromatography does not represent the bulk sample, account for volatiles or solid masses in the reaction mixture, etc. and can over or underestimate RCY. We have now removed RCY from all Tables/Schemes unless the product is isolated.

While our goal is ultimately for PET tracer production, the aim of this Communication is to develop a new chemical method. However, in light of the Reviewer's comment, we have carried out additional experiments to align with the precise definition of RCY, and have now manually synthesized [^{18}F]**2a** in 2% RCY, as proof of concept with a small amount of starting radioactivity. The molar activity (60.2 mCi/nmol) was determined for [^{18}F]KP23 and is low because of the starting activity and based on ALARA principles we cannot ethically justify a scaled up manual reaction at this time. These sections have been added to the radiochemistry section of the revised manuscript and SI.

Response to the Reviewer 3's comments

In this work, the authors described a FePc-catalyzed aminofluorination of alkenes, providing a valuable route to N-unprotected β -fluoroamines. Although there have been some pioneering reports on alkene aminofluorination reactions (ref. 26-43), and studies on the use of hydroxylamine reagents as NH_2 radical sources in alkene difunctionalizations (ref. 50-62), this work remains noteworthy, as it offers an efficient method for preparing synthetically valuable β -fluoroamines, without the need for deprotection steps. The application in radiosynthesis further highlighted the practical utility of this work. The authors also demonstrate a range of synthetic transformations of the reaction products, including the synthesis of several pharmaceutically relevant molecules. Mechanistic studies, including DFT calculations, were performed, revealing a potential catalytic pathway for this reaction. Overall, the manuscript is well-organized, and I believe it would be of interest to the readership of Nat. Comm., after addressing the following problems/issues:

Response: We thank the Reviewer for the positive evaluation and strong support of this work.

1) The authors state that unactivated alkenes are compatible with this reaction, yet all the examples in Figure 2 are substituted styrenes or 1,1-disubstituted alkenes. The reactivity of

monosubstituted alkenes such as 1-octene should be discussed, and an example should be provided if possible.

Response: According to the suggestion of the Reviewer, we have employed 1-octene as the substrate. Unfortunately, the reaction failed probably due to the slow addition of *N*-centered radical to the π system of mono-substituted unactivated alkene, which led to the decomposition of highly active radical species. About 80% alkene substrate was recovered. This information has been added in the revised main text and Figure 2.

2) In DFT calculations, the oxidation state of Fe in INT4 is described as +4, which is somewhat confusing, as such a formal +4 oxidation state is typically exhibited only in iron-nitrenoid species. A more precise description of this species should be included in the manuscript. Additionally, is it possible for the reaction to proceed directly from INT3 without undergoing INT4?

Response: The oxidation state of iron was assigned based on the electronic configuration. As shown in Figure R1, the electronic configuration of INT indicates that the iron center has 4 d-electrons (blue lines). Thus, the oxidation state of Fe in INT4 is described as +4. Similar ferryl amino species were reported in literatures, *e.g.* *ACS Catal.* **2023**, *13*, 1863.

The reaction is unlikely proceeds directly from INT3 without undergoing INT4. The attempts to locate the transition state from INT3 always lead to the same TS4 that features very little spin population ($\rho_{Pc} = -0.02$) on the Pc ligand, suggesting significant change in the electronic structures occurred for INT3 to approach the TS structure. The intrinsic reaction coordinate (IRC) optimizations of TS4 lead to a ferryl amino intermediate INT4. Relative to INT3, INT4 represents an iron to Pc charge-transfer excited state. Such electronic structure reconfiguration increases the N component and lowers the energy level of the electron-accepting molecular orbitals ($\sigma_{z^2}^* \alpha$: 9% N \rightarrow 23% N, $-2.14 \rightarrow -4.04$ eV; $\pi_{yz}^* \beta$: 20% N \rightarrow 24% N, $-2.14 \rightarrow -3.95$ eV), thereby increasing the electrophilicity of the intermediates and enhancing its reactivity towards the alkene.

Figure R1. Electronic configuration, orbital components and energy levels of electron-accepting MOs of INT3 and INT4. Fe, Pc, and nitrogen-based molecular orbitals are shown and denoted as blue, black, and

gray lines, respectively. Spin populations (ρ) are given in atomic units (a.u.).

3) In Figure 5, is there a hydrogen bond between aziridinium and FePc in INT6? The structural formula and description are unclear and may be misleading. Moreover, another potential mechanistic pathway should be considered, in which FePc has already dissociated in INT6, and TS6 corresponds to a nucleophilic addition of $\text{Et}_3\text{N}\cdot 3\text{HF}$ to the aziridinium without Fe-catalysis. Additionally, I could not find the Cartesian coordinate of TS6 in the SI, and it seems that the author may have mixed up the numbering. A careful review of the SI is necessary.

Response: Yes, there is a hydrogen bond between aziridinium and FePc in INT6 (Figure R2a). We have modified the structural formula to explicitly show the hydrogen bond.

According to the suggestion of the Reviewer, we calculated the reaction pathway in which FePc has already dissociated in INT6 (Figure R2b). Our computational results suggest that the FePc dissociation process is slightly endothermic by 0.8 kcal/mol and the nucleophilic addition of $\text{Et}_3\text{N}\cdot 3\text{HF}$ to the free aziridinium is 2.5 kcal/mol less favorable than that with the assistance of FePc. We have carefully examined and corrected the labels in the revised SI.

Figure R2. (a) Optimized 3D structure of INT6. (b) Reaction profile for the nucleophilic addition of $\text{Et}_3\text{N}\cdot 3\text{HF}$ in the presence and absence of FePc.

4. Some typos:

Line 138, respectively -> respectively

Line 200, source -> source

Response: We have made the revision according to the suggestion.

Response to the Reviewer 2's Comments

The authors have shown that they can obtain amines directly for a large number of derivatives. However, I still find Figure 2 problematic in terms of the molecule numbering. It is not acceptable to display molecules in this figure with a “2x” number and an associated yield, represented as amines, while in the Supporting Information these same molecules are in fact carbamates with the same yield. I find the footnote system used in Figure 2 (a and b) by the authors to be lacking in rigor, and I believe this needs to be corrected. The same issue appears in Figure 6 for the radiolabeled products.

Response: We thank the Reviewer for this suggestion. The footnote system of Figure 2 has been revised accordingly; superscript 'a' now indicates the yield of the isolated product after Boc protection.

The authors have clarified the misunderstandings regarding RCC and RCY and have added two molar activities. However, they should report the specific activities using a more conventional unit, GBq/μmol for example. While the reported values are good, using a more appropriate unit would allow readers to understand them at a glance (EJNMMI Radiopharm. Chem. 6, 34 (2021). <https://doi.org/10.1186/s41181-021-00149-6>). Additionally, the authors should, in my opinion, include in the Supporting Information the data used to determine the reported molar activities (calibration curve, total and isolated activities....). I believe this is consistent with current standards in radiochemistry (J. Am. Chem. Soc. 2024, 146, 11599–11604).

Response: The manuscript has now been updated with the molar activities represented in GBq/nmol. The Supporting Information has also been updated with the calibration curves (Supplementary Fig. 24 and 31) and relevant molar activity information (Supplementary Tables 4 and 5).